# Motion Control System of Unmanned Railcars Based on Image Recognition

**Yuan-Wei Tseng** **, Tsung-Wui Hung, Chung-Long Pan and Rong-Ching Wu \***

Department of Electrical Engineering, I-Shou University, Kaohsiung City 84001, Taiwan;
yuanwei@isu.edu.tw (Y.-W.T.); z5151381@gmail.com (T.-W.H.); ptl@isu.edu.tw (C.-L.P.)

**\*** Correspondence: rcwu@isu.edu.tw

**Abstract:** The main purpose of this paper is to construct an autopilot system for unmanned railcars based on computer vision technology in a fixed luminous environment. Four graphic predefined signs of different colors and shapes serve as motion commands of acceleration, deceleration, reverse and stop for the motion control system of railcars based on image recognition. The predefined signs' strong classifiers were trained based on Haar-like feature training and AdaBoosting from Open Source Computer Vision Library (OpenCV). Comprehensive system integrations such as hardware, device drives, protocols, an application program in Python and man machine interface have been properly done. The objectives of this research include: (1) Verifying the feasibility of graphic predefined signs serving as commands of a motion control system of railcars with computer vision through experiments; (2) Providing reliable solutions for motion control of unmanned railcars, based on image recognition at affordable cost. The experiment results successfully verify the proposed methodology and integrated system. In the main program, every predefined sign must be detected at least three times in consecutive images within 0.2 s before the system confirms the detection. This digital filter like feature can filter out false detections and make the correct rate of detections close to 100%. After detecting a predefined sign, it was observed that the system could generate new motion commands to drive the railcars within 0.3 s. Therefore, both real time performance and the precision of the system are good. Since the sensing and control devices of the proposed system consist of computer, camera and predefined signs only, both the implementation and maintenance costs are very low. In addition, the proposed system is immune to electromagnetic interference, so it is ideal to merge into popular radio Communication Based Train Control (CBTC) systems in railways to improve the safety of operations.

**Keywords:** image recognition; computer vision; OpenCV; AdaBoosting; motion control

---

## 1. Introduction

In many automation applications, for example, sushi trains in restaurants, automatic manufacture lines, warehouse storages and Mass Rapid Transit (MRT), unmanned self-propelled railcars play many important roles. Many railcars operate in surroundings of fixed ambient brightness, such as indoor or subways.

Unmanned railcars should be able to control their own motions in response to different situations automatically. Due to their functional similarity, the developing control systems for unmanned railcars can borrow some distinguishing features from more complicated train control systems in a railway. Therefore, we first briefly review the evolution of the train control system. Basically, the train control system in a railway is a signaling system. If a train can correctly identify the signals received from tracksides, it can operate safely.

In the early days of railways, when train speed was not fast, locomotive engineers operated trains by watching signal lights and signs on the tracksides, and then manually made proper decisions to control the motions of trains without colliding with other trains. Later on, when train speeds were too fast for locomotive engineers to correctly recognize trackside signals and operate trains safely, Track Circuit Based Train Control systems (TBTC) [1] were developed to ensure the safety of railway operations. A track circuit is a simple electrical device used to detect the absence of a train on rail tracks, and thus to inform signalers and control relevant signals as shown in [2].

Nowadays, with the popularity of high-speed railways and the demand of increasing capacity for more rapid travel with more passengers, the maintenance time left for track circuits becomes less and less. To overcome this situation, the more advanced Communication Based Train Control (CBTC) system [1,3–5], whose system overview as shown in [6] was introduced and became the modern railway signaling system for mass rapid transit systems worldwide. A CBTC system is a "continuous, automatic train control system utilizing high-resolution train location determination, independent from track circuits; continuous, high-capacity, bidirectional train-to-trackside data communications; and train-borne and wayside processors capable of implementing Automatic Train Protection (ATP) functions, as well as optional Automatic Train Operation (ATO) [5] and Automatic Train Supervision (ATS) functions", as defined in the IEEE 1474 standard [7].

Currently, CBTC systems apply radio devices to provide continuous and high-speed data communications such as monitoring train speeds and track conditions between train and tracksides. Since the CBTC systems can measure train speeds, the optimal design of CBTC ATO speed profiles that minimizes the energy consumption for a target running time, were studied [8,9]. From our above brief introduction, we know that both the TBTC system and CBTC system have complex hardware architectures. There are many detecting devices distributed along the tracksides. Therefore, both implementation cost and maintenance cost are very high. Furthermore, equipment malfunction, weak signal strength, saturation of the communications medium or electromagnetic interference can result in failure of the train control system, thus affecting normal train service. Detecting and fixing the failure could be a time consuming and costly task. For these reasons, an auxiliary control system of CBTC system is desirable.

Unlike the trains in railway systems that transport both passengers and cargos on different spurs, railcars usually have their own dedicated function and travel on a fixed path. Therefore, railcars usually have less safety concern and are more suitable for unmanned operations. For simple applications, TBTC as well as CBTC is too complex and too costly for controlling the motions of railcars.

According to briefly reviewing the history of signaling systems of train control, we find that one thing in common in the evolution of train signaling systems. That is the data communications between train and trackside devices. Inspired by this idea, and considering that many railcars actually operate in fixed luminous environments, we developed a motion control system of unmanned railcars based on image recognition in this research. Although train drivers cannot correctly recognize signs and signals along the tracksides in high-speed railways, computer vision can.

The idea is first to select four predefined signs of different colors and shapes serving as basic motion commands of acceleration, deceleration, reverse and stop for the railcar motion control system. Directly working with only image intensities (i.e., the RGB pixel values (red, green and blue light are added together in various ways to reproduce a broad array of colors) of image) made the task of feature calculation time consuming and computationally expensive. In our application, the railcar must be able to identify four predefined signs while it is moving, and the applied image recognition algorithm should be fast and accurate enough.

Paul Viola and Michael Jones developed a very successful fast face detection system [10]. In that system, they prominently used integral images and adapted the idea of using Haar wavelets to develop the so-called Haar-like features to train strength classifiers based on AdaBoosting and Cascade algorithms for detecting faces. Lienhart et al. [11] further proposed an extended set of Haar-like features to get further improvement.

Paul Viola and Michael Jones' stronger classifier often outperforms most 'monolithic' strong classifiers, such as support vector machines and Neural Networks [12]. Image recognition based on AdaBoosting and Cascade algorithms were applied in autonomous vehicle applications [13–16]. In those applications, other than camera, Light Detection and Ranging (LIDAR) [13], Global Positioning System (GPS) [14–16], and road surface maker (RSM) [15,16] databases in the cloud are also used. In our application, other than a camera, we do not need additional sensing devices.

In this research, Viola and Jones' AdaBoosting and Cascade algorithms [10] together with Lienhart's extended set of Haar-like features [11] are adopted to train the four predefined signs. The control system is pre-trained and is able to recognize those predefined signs in consecutive image frames. We then can properly distribute those predefined signs along the tracksides according to kinematics pre-analyses. When the onboard camera of the railcar transfers continuous image frames to the host computer, the computer can detect those predefined signs and respond properly to control the motion of the railcars.

In addition, if we select a trained predefined sign as a positioning sign, we can distribute them with equal distance along the trackside, dividing the distance by the time difference of two consecutive positioning sign detections, and as such, the system can get the estimation of railcar speed just like the CBTC system. With this concept, we constructed an image recognition based motion control system of railcars with extremely low maintenance and immunity against electromagnetic interference in this research. This system can also serve as an auxiliary speed measurement system for the CBTC system, which easily suffers from electromagnetic interference. Open Source Computer Vision Library (OpenCV) [17] is a library for real time computer vision provided by Intel, and OpenCV utilities are properly applied in developing the integrated programs for the system. The operations of the proposed motion control system of railcars are as follows.

The on board camera of the railcar serves as the sensor. It transfers continuous image frames through a wireless microwave receiver to the host computer. This host computer identifies the four predefined signs and then generates the proper pulse width modulation (PWM) commands [18] to control the motions of the railcar. The whole control system is compact, low cost and easy to maintain. The railcars used in this experiment consist of a powered car and a camera on board the car. They are DC voltage controlled mini train models with scale of 1/150 [19]. When more predefined signs are trained, the experiment setup can be modified to accomplish more complex control tasks for railcars. To the author's best knowledge, no similar railcar motion control system was reported in the literature.

## 2. Detecting Predefined Signs with Image Recognition

The most crucial part of the proposed motion control system of railcars is how to detect predefined signs serving as motion commands successfully. In this section, the methods and procedure to detect predefined signs are provided and explained as follows.

### 2.1. Image Pre-Processing Using OpenCV

OpenCV library [17] provides many useful utilities for image processing and is very popular in the industry. For development effectiveness and maintenance, in this research a variety of OpenCV utilities are applied in main program so that the designed system is able to quickly and correctly identify four different predefined signs from the images captured by a Metal–Oxide–Semiconductor (CMOS) camera. Careful consideration of image pre-processing is fundamental for computer vision applications because raw image data direct from a camera may be noisy and unevenly lighted. Therefore, proper image pre-process tasks should be performed to handle the noisy and unevenly lighted situations.

In practical, like the application of this research, many object features in image are brightness, contrast, edges, shape, contours, texture, perspective or shadows related, rather than color related. To increase the speed of particular object detection, the captured colorful RGB images are converted to gray level images because the data volume of a gray level image with 8-bit resolution is significantly less than that of its original colorful RGB image. Therefore, the first step of image processing is to

convert the captured color images to gray level images. The conversion between color RGB images to gray level Y images is based on the formula in (1) [20]:

$$Y = 0.299R + 0.587G + 0.114B. \tag{1}$$

In this research, the image recognition and motion control program was developed in Python [21]. The images captured by the camera were applied OpenCV function *cv2.cvtColor* [22] to convert the input color RGB images to gray level images.

Secondly, since the captured images are usually noisy, we applied OpenCV function *cv2.GaussianBlur* [23], which convolves the gray level images with the specified Gaussian kernel to reduce the image noises.

Thirdly, in gray level images, image features usually relate to image contrast. When contrast values are very close, the features in images are difficult to detect. Histogram equalization is a method in image processing of contrast adjustment using the image's histogram. Through this adjustment, the intensities are better distributed on the histogram. This allows for areas of lower local contrast to gain a higher contrast.

Histogram equalization accomplishes this by effectively spreading out the most frequent intensity values. In other words, this algorithm normalizes the brightness and increases the contrast of the image. The *cv2.equalizeHist* [24] is the dedicated function to handle the histogram equalization of gray level images in OpenCV.

## 2.2. Image Recognition of Predefined Signs

After image preprocessing, the system needs to detect the predefined signs that serve as motion commands in the consecutive gray level image frames. In this research, we need four strong classifiers to detect the four predefined signs of acceleration, deceleration, reverse and stop in consecutive gray level image frames. The system first extracts Haar-like features from consecutive gray level image frames. Then, input those features to predefined signs' classifiers in the main program. Once a predefined sign is detected, the system can respond properly to generate PWM commands to control the motion of the railcar.

To enhance the integrity of this paper, how the integral image based extended set of Haar-like features works with the AdaBoost algorithm [10–12,20], and can form a strong classifier to identify a predefined sign in image frames, is briefly explained in this section.

For a gray level image, an integral image of a pixel with coordinates $(x, y)$ is the sum of its neighbors to the upper left, and is defined in (2):

$$ii(x,y) = \sum_{x' \leq x} \sum_{y' \leq y} I(x',y'). \tag{2}$$

Mathematically speaking, unlike the original pixels' gray level values that have many identical numbers in an image frame, their integral image values of pixels are quite different from each other. Therefore, integral image is very useful for unique feature extractions (See Figure 1).

$$ii_1 = Sum(A), \tag{3}$$

$$ii_2 = Sum(A) + Sum(B), \tag{4}$$

$$ii_3 = Sum(A) + Sum(C), \tag{5}$$

$$ii_4 = Sum(A) + Sum(B) + Sum(C) + Sum(D), \tag{6}$$

$$Sum(D) = ii_1 + ii_4 - (ii_2 + ii_3), \tag{7}$$

$$Sum(C) = ii_3 - ii_1, \tag{8}$$

$$Sum(B) = ii_2 - ii_1. \tag{9}$$

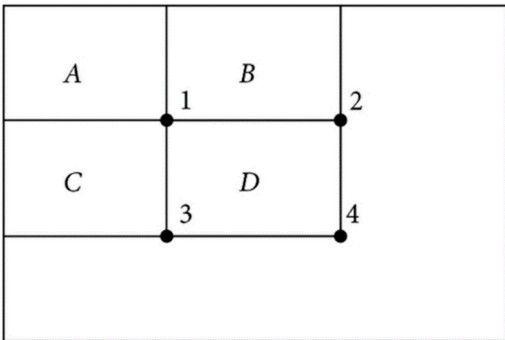

**Figure 1.** Integral image blocks.

According to Figure 1, applying (3)–(9), one can compute the integral images of individual blocks. Based on these integral images of individual blocks, the extended Haar-like rectangle features with different black and white areas as shown in the Figure 2 of [11] are used. They are sensitive to the presence of edges, bars and other simple image structures. In those features, black areas have negative and white areas positive weights. Two-rectangle feature is the sum of the pixels, which lie within the white rectangles being subtracted from the sum of pixels in the black rectangles, while a three-rectangle feature computes the sum within two outside rectangles, subtracted from the sum in a center rectangle. The number of features derived from extended Haar-like rectangle features is quite large.

For example, for a 24 × 24 detection window, the total number of features is 117,941—that is a number far larger than the number of pixels. Please refer to [11] for details. Actually, only very small numbers of these features are enough to combine to form an effective strong classifier. In this paper, AdaBoost algorithm [20,25] is then employed to select the most suitable features.

AdaBoost, short for Adaptive Boosting, is an iterative method to improve the accuracy of machine learning. Boosting is an ensemble technique (a.k.a. a committee method). A weak classifier with the least error rate is selected in every iteration. When the iterations finish, a strong classifier of highly accurate prediction is created by combining all weak learners. In this paper, the AdaBoost algorithm is chosen to build the four strong classifiers for both high accuracy and fast speed.

To train a strong classifier with this AdaBoost algorithm, we have to prepare $m$ training sets $\langle x_i, y_i \rangle$, where $x_i$ is the ith sample, $y_i$ is the corresponding label of $x_i$, and $1 \le i \le m$. For our application, the sample $x_i$ is a vector of Haar-like features extracted from a training image frame. The label $y_i = +1$ indicates the predefined sign shows up in the training image and the corresponding sample $x_i$ is a positive sample. On the other hand, $y_i = -1$ indicates the predefined sign does not show up in the training image and the corresponding sample $x_i$ is a negative sample. Therefore, $m$ training sets contain both positive and negative samples and their corresponding labels.

We should then set the training iteration number $K$, and assign the initial weight $W_i^0 = \frac{1}{m}$ to every feature vector in the training sets. Collecting the jth element from every n × 1 Haar-like feature vector $x_i$ in the $m$ training set to form $n$ new sets $\Gamma_j$ described in (10) as follows:

$$\Gamma_j = \left\{ x_{i(j)} \middle| \forall i \in \{1, 2, \ldots, m\}, \, j \in \{1, 2, \ldots, n\} \right\}. \tag{10}$$

Sorting the elements of $\Gamma_j$ according to numeric values, from left to right and from the smallest to the greatest, we have another $n$ new sets $V_j$, which have sorted elements in (11):

$$V_j = \left\langle v_{1(j)}, v_{2(j)}, \ldots, v_{m(j)} \right\rangle. \tag{11}$$

The threshold value candidate set for the jth element of Haar-like feature vectors is calculated by

$$T_{V_j} = \left\{ t_s = \frac{v_{s(j)} + v_{s+1(j)}}{2} \middle| \forall s \in \{1, 2, \dots, m-1\}, j \in \{1, 2, \dots, n\} \right\}. \tag{12}$$

Basically, $T_{V_j}$ in (12) is formed by taking the averages of every two adjacent elements in $V_j$ in (11). Therefore, $T_{V_j}$ sets consist of $n \times (m-1)$ threshold value candidates for weak classifiers.

In the kth iteration, every element in $\Gamma_j$ goes through both greater than and smaller than comparisons with threshold candidates $t_s$ in (12). The comparison results $R^k_{t_s, \Gamma_j}$ are given as

$$R^k_{t_s, \Gamma_j} = \langle h^k_{t_s} \left( x_{i(j)} \right) | \forall x_{i(j)} \in \Gamma_j, \forall x \in \{1, 2, \dots, m\}, j \in \{1, 2, \dots, n\} \rangle. \tag{13}$$

In (13), $h^k_{t_s} \left( x_{i(j)} \right)$ represents weak classifier candidates.

$$h^k_{t_s} \left( x_{i(j)} \right) = \begin{cases} 1, & if \ p x_{i(j)} < p t_s \\ -1, & otherwise \end{cases} \tag{14}$$

In (14), each weak classifier candidate has two associated variables. They are threshold $t_s$ and polarity $p$. Polarity $p = 1$ for the classifier candidates in which $x_{i(j)} < t_s$, while polarity $p = -1$ for the classifier candidates in which $x_{i(j)} > t_s$. If the result of the weak classifier $h^k_{t_s} \left( x_{i(j)} \right)$ in (14) is the same as the corresponding label $y_i$ in the training set of $\langle x_i, y_i \rangle$; this weak classifier works correctly for $x_i$; otherwise, it does not, and error occurs. In every training iteration, the error rates $\phi^k_j$ of classifier candidates are calculated with the following equations in (15)–(17).

$$\omega^k_i \left( R^k_{t_s, \Gamma_j} \right) = \begin{cases} W^k_i, & if \ h^k_{t_s} \left( x_{i(j)} \right) \neq y_i \\ 0, & otherwise \end{cases}, \tag{15}$$

$$e^k_s \left( R^k_{t_s, \Gamma_j} \right) = \sum_{i=1}^{m} \omega^k_i \left( R^k_{t_s, \Gamma_j} \right), \tag{16}$$

$$\phi^k_j = \left\{ e^k_s \left( R^k_{t_s, \Gamma_j} \right) \middle| \forall s \in \{1, 2, \dots, m-1\}, \forall t_s \in \Gamma_{V_j} \right\}. \tag{17}$$

The best classifier candidate $h^k_{t^k_{o(\eta^k)}} \left( x_{i(\eta^k)} \right)$, whose polarity $p^k$ and threshold $t^k_o$ for the $\eta^{k\text{th}}$ element of Haar-like feature vectors, has the smallest error rate $\varnothing^k$, and is selected as a weak classifier in the kth iteration.

In the kth iteration, the voting weight $\rho^k$ of the selected weak classifier $h^k_{t^k_{o(\eta^k)}} \left( x_{i(\eta^k)} \right)$ in the final strong classifier is calculated based on the $\varnothing^k$ and given in (18):

$$\rho^k = \frac{1}{2} ln \left[ \frac{\left( 1 - \phi^k \right)}{\phi^k} \right]. \tag{18}$$

The correct rate of the selected weak classifier should be higher than 0.5, which is the probability of guess. Therefore, the smallest error rate $\varnothing^k$ should be small than 0.5. Consequently, $\rho^k$ is positive, and the smaller the $\varnothing^k$, the bigger the $\rho^k$. In other words, the weak classifiers with smaller error rates are more important in the final strong classifier.

In every iteration, the weight of $m$ feature vectors are updated according to $\rho^k$. If the normalization factor $Z^k$, namely, the summation of weights of feature vectors is defined as follows:

$$Z^k = \sum_{i=1}^{m} W^k_i. \tag{19}$$

According to (19), the normalized weight updates are

$$W_i^{k+1} = \frac{W_i^k}{Z^k} \times \begin{cases} exp(-\rho^k), \ if \ h_{t_{o(\eta^k)}^k}^k \left( x_{i(\eta^k)} \right) = y_i \\ exp(\rho^k), \ if \ h_{t_{o(\eta^k)}^k}^k \left( x_{i(\eta^k)} \right) \neq y_i \end{cases}. \tag{20}$$

Therefore, from (20), we can find that in next training iteration, the weights of correct classifiers are decreased, while the weights of incorrect classifiers are increased. Training samples misclassified by a previous weak learner are given more emphasis at future rounds. After *K* training iterations are done, all obtained weak classifiers multiplied by their own voting weight $\rho^k$ are combined to form the following final strong classifier based on a weighted majority voting method.

$$\text{H}(x_i) = sign\left[ \sum_{k=1}^{K} \rho^k h_{t_{o(\eta^k)}^k}^k \left( x_{i(\eta^k)} \right) \right] \tag{21}$$

Therefore, the final strong classifier in (21) is a combination of the weak ones, weighted according to any error they had. Four predefined signs, namely, acceleration, deceleration, reverse and stop, need their own strong classifiers. Therefore, we have to train four strong classifiers for them.

### 2.3. Designs of Predefined Signs Serving as Motion Commands

The designed predefined signs serving as motion commands are shown in Figure 2.

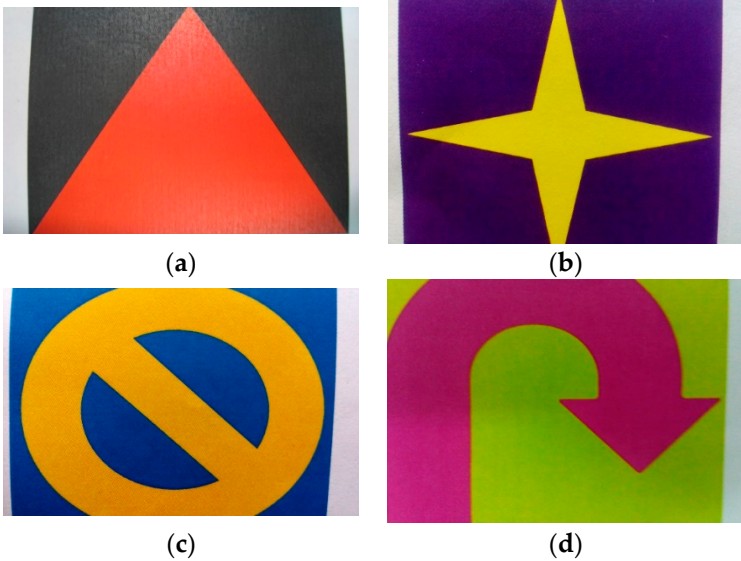

(**a**)         (**b**)

(**c**)         (**d**)

**Figure 2.** Predefined signs serving as motion commands: (**a**) acceleration; (**b**) deceleration; (**c**) stop; (**d**) reverse.

The predefined signs serving as motion commands should be designed, such that the system can detect them easily and correctly. However, humans do not need to recognize them. Therefore, predefined signs should have good geometric and color features, so that the system can quickly distinguish them. In Figure 2, we selected the four predefined signs with different geometric shapes and color contrasts to serve as motion commands of acceleration, deceleration, reverse and stop. The experimental results with respect to detecting those signs were good.

### 2.4. Training Strong Classifiers of Motion Comands

Since the railcars used in this experiment are mini train models with a scale of 1/150, the size of the predefined signs used in the experiment is only 1 cm by 1 cm. However, when training the strong

classifiers for those signs, the photo samples were enlarged to 9 cm × 9 cm to retain much more detail, and avoid distortion so that we can get better results. Using the acceleration sign as our example, to train the strong classifier we have to prepare two image sample folders. One is the positive sample folder, which contains photos of our acceleration sign taken at slightly different angles, as shown in Figure 3.

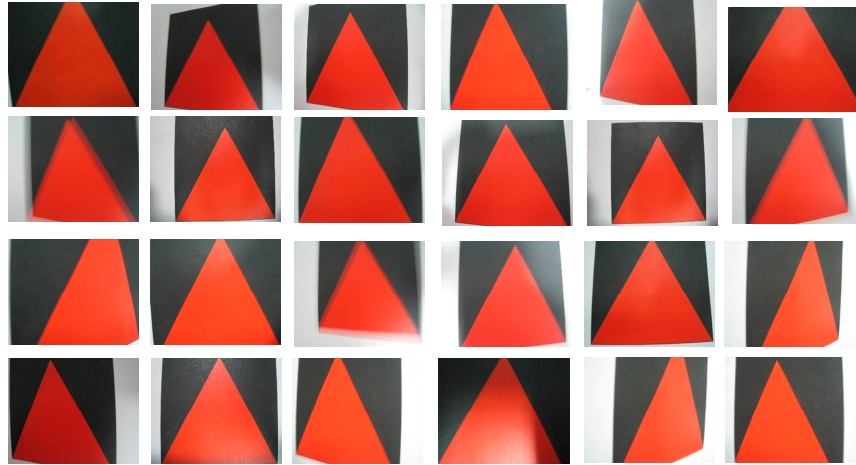

**Figure 3.** Some positive samples of acceleration sign.

The other is the negative sample folder of background sample images, in which every photo has no acceleration sign as shown in Figure 4.

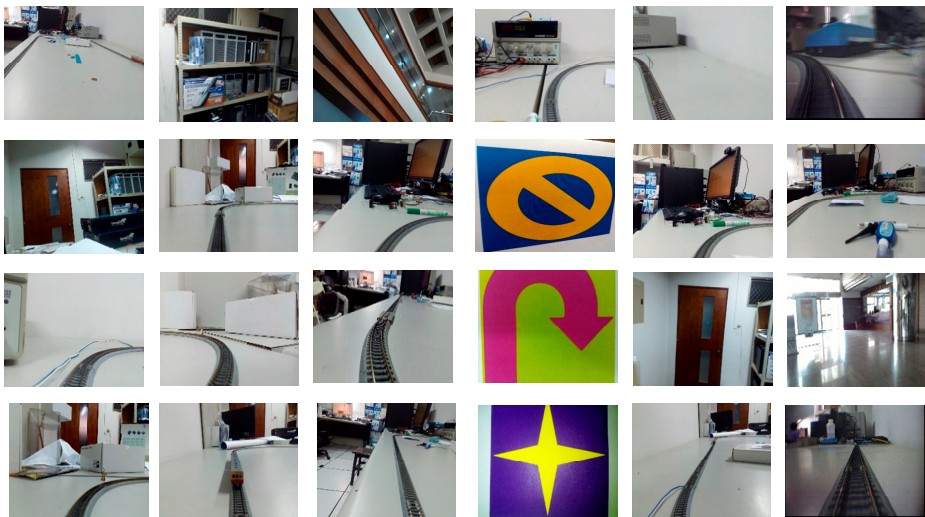

**Figure 4.** Some negative samples of acceleration sign.

OpenCV (Open Source Computer Vision Library) [17] provides many image processing related functions and utilities. OpenCV has adopted Viola and Jones' methodology and Lienhart's extended set of Haar-like features to provide two very useful utilities, *opencv_createsamples* and *opencv_traincascade* [26] for training strong classifiers based on AdaBoost. To apply them, negative samples are further enumerated in a background description file [26]. It is a text file, in which each line contains an image filename (relative to the directory of the description file) of a negative sample image. With the prepared positive sample images and background description file, we first follow the instructions of applying *opencv_createsamples* to generate an output vector file containing the positive samples for training. With the output vector file containing the positive samples obtained from *opencv_createsamples* and the background description file, we then follow the instructions of using

*opencv_traincascade* to obtain the strong classifier of acceleration in xml format. Repeating the same process we can obtain strong classifiers of deceleration, reverse and stop signs. After a classifier of a predefined sign is trained, it can be applied to a region of interest (of the same size as that which was used during the training) in an input image. The classifier outputs a "1" if the region is likely to show the predefined sign, and "−1" otherwise. To search for the predefined sign in the whole image, one can move the search window across the image and check every location using the classifier. The classifier is designed so that it can be easily "resized" in order to be able to find the predefined sign at different sizes, which is more efficient than resizing the image itself. Therefore, to find a predefined sign of an unknown size in the image, the scan procedure should be done several times at different scales. This can be done by applying *CascadeClassifier::detectMultiScale* [27] in OpenCV.

In experiment, the accuracy of the strong classifier related to $K$, the Number of iterations or cascade stages, should be trained. If $K$ is too small (under training), false detections of non-predefined signs happen. On the other hand, if $K$ is too large (over training), predefined signs could be overlooked by the system. Therefore, we should adjust $K$ to get the classifiers with high accuracy.

## 3. Experiment System Architecture and Setting

The integrated system architecture is given in Figure 5. From the bottom, the hardware layer consists of the sensing devices, actuating devices and a PC as the controller. The sensing device is a CMOS camera with a resolution of 270,000 pixels installed on the N scale (1/150) model railcar, which transmits the captured analog images to our main computer through wireless microwave.

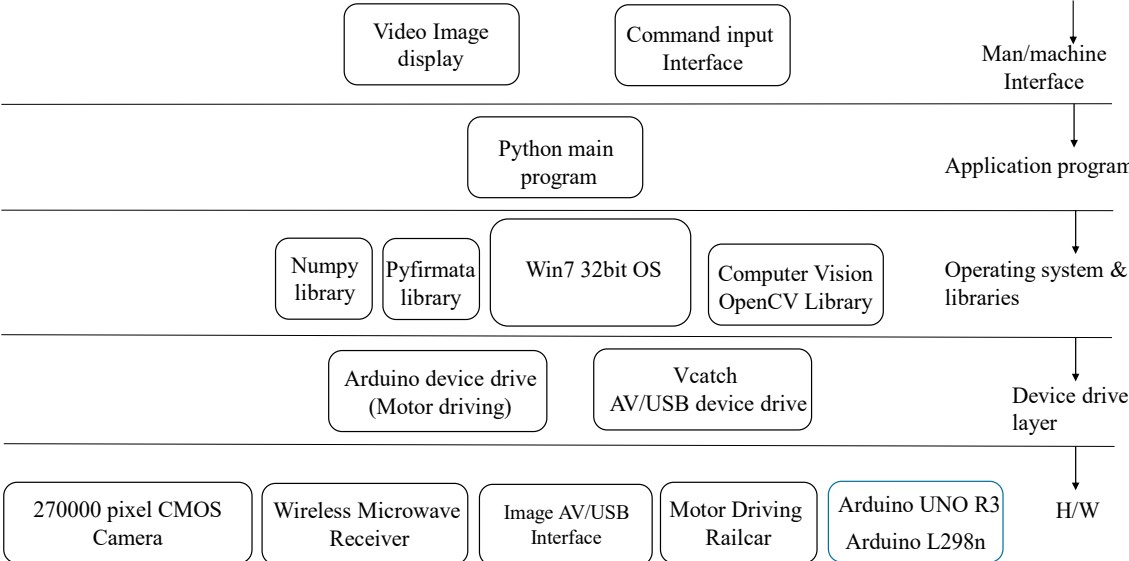

**Figure 5.** Integrated system architecture.

The analog images received by wireless this microwave receiver are converted to digital images through an image converter that is connected to the controller PC with a USB interface. The PC processes the consecutive input image frames, and then properly sends motion commands to the actuating device, this being the Arduino UNO R3 board (Arduino, Somerville, MA, USA) [28] with L298n motor driving module to control the motion of the N scale (1/150) model railcars. Therefore, the device drive programs of both the Vcatch image converter and the Arduino board should be included in the device drive layer of the system. The PC controller uses the Windows 7 32-bit operating system. Numpy [29] is the fundamental package for scientific computing with Python [17], and is imported into the system. Firmata [30] is a protocol for our host computer to communicate with the Arduino UNO R3 microcontroller, while pyFirmata is the Python interface of the Firmata protocol. Both of them are applied in the system. OpenCV (Open Source Computer Vision Library) [17] is an open source

computer vision and machine learning software library. In this paper, image processing and computer vision related tasks use many utilities provided by the OpenCV library. Therefore, the Numpy library, Pyfirmata library, and OpenCV library, together with the Windows 7 32-bit operating system, build the operating system and library layer. The application main program is written in Python. The system provides a video display interface in order to display the images captured by the CMOS camera (RF System lab, Nagano, Japan) installed on the model railcar, such that the user may visualize what is seen by CMOS camera. There is also a man/machine interface for the user to control the motion of the model railcar by entering the proper commands manually and remotely. In this research, we used 32-bit Python 2.7.11, OpenCV2.4.11, Numpy, Arduino1.6.9 and Pyfirmata in our control system.

The experiment facility is shown in Figure 6.

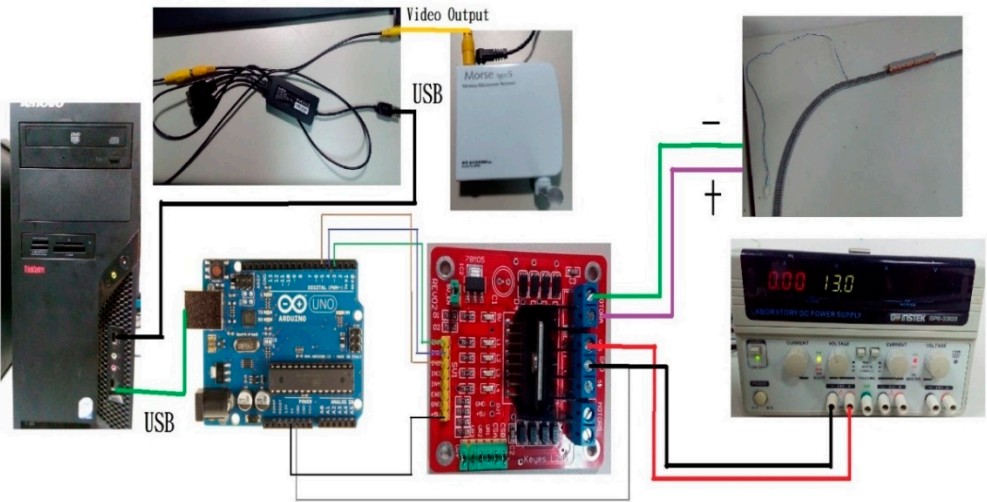

**Figure 6.** The experiment facility.

## 4. Experiment Results

### 4.1. Introduction to Testing Program

After setting up our experiment facility, the test program was run to verify the proposed method. When the program starts to execute, the model railcar is given 55% of its power to run smoothly. At the same time, an input interface window, as shown in Figure 7, jumps out onto the screen. The input interface can take the commands of four different users, these being "a", "d", "s" and "t", where "a" stands for accelerate, "d" for decelerate, "s" for stop and "t" for turn around.

**Figure 7.** Input interface window.

When the model railcar is running, its wireless CMOS camera starts to send captured consecutive images to the PC. Meanwhile, a window on the PC screen as shown in Figure 8 will jump out to display the consecutive images being received from the wireless CMOS camera.

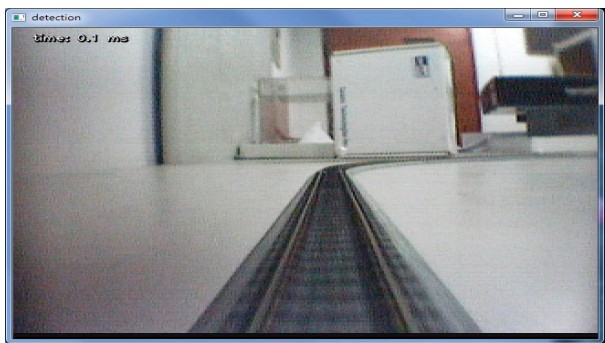

**Figure 8.** Images from the wireless CMOS camera.

The received consecutive images are pre-processed using utilities provided by OpenCV. The images are first converted to a gray level image using cv2.cvtColor. Secondly, applying a Gaussian filter using cv2.GaussianBlur will reduce the noise of these gray level images. The function convolves the source image with the specified Gaussian kernel. In the third step, the cv2.equalizeHist function is utilized to equalize the histogram of grayscale images so that the levels of brightness are normalized and the contrast of the images are increased.

Since the accuracies of the trained strong classifiers are not 100%, all four predefined signs, these being, accelerating, decelerating, stopping and reversing, should be detected at least three times in consecutive images within 0.2 of a second before the system confirms the detections. This method can filter out any false detection, which just acts like a glitch. As shown in Figure 9, when the detections are confirmed, the detected signs will be enclosed by different color rectangles with proper text notations to describe which predefined sign it is.

Almost simultaneously, the motion PWM command corresponding to the detected predefined sign is transferred to the Arduino UNO R3 board and its L298n motor driving module in order to control the motion of the N scale (1/150) model railcar. For the model railcar, if the given power is over 95%, it could run too fast and go off the rails. Therefore, the allowed maximum power in the program is set at 95%. On the other hand, if the power is below 35%, the model railcar cannot go further. The required minimum power in the program is set at 35% Every time when an acceleration sign is detected, the power is increased by 10% until the maximum of allowed power, which is 95%, is reached, to ensure that the model railcar will not go off the rails. Similarly, every time that a deceleration sign is detected, the power is decreased by 10% until the minimum of required power of 35% is reached.

When a stop sign is detected, the power immediately drops to zero to stop the model railcar. When a reverse sign is detected, program will reverse the model railcar's moving direction by setting digital inputs IN1 to zero, and IN2 to one on the Arduino L298n motor driving module so that the model railcar can immediately reverse, and also increase its power by 10% to quickly move away from its original direction.

In the experiment, we deployed four predefined signs to do the full functional verification. The railcar first detected the acceleration sign and it did accelerate accordingly. Then it detected the deceleration sign, and the speed was slowed down as expected. Followed by its detection of the reverse sign, the train shortly stopped and turned around to move in the opposite direction. When the railcar was moving in the opposite direction, the camera was in its tail. When the tail camera detected both deceleration and acceleration signs again, the railcar had according responses. It kept moving until the camera detected a stop sign and the railcars immediately stopped. The camera perspective film of full functional experiment is available at https://www.youtube.com/watch?v=6QnGb2CMvwc

(accessed on 31 January 2019) [31]. The bystander perspective film of full functional experiment is available at https://www.youtube.com/watch?v=TZ-kg4dFVjM (accessed on 31 January 2019) [32].

Therefore, experimentation successfully verified the proposed method of unmanned railcar motion control based on real-time image recognition.

With analyses of kinematics, we can properly deploy the motion command signs of acceleration, deceleration, reverse and stop along the trackside to control the motion of railcars.

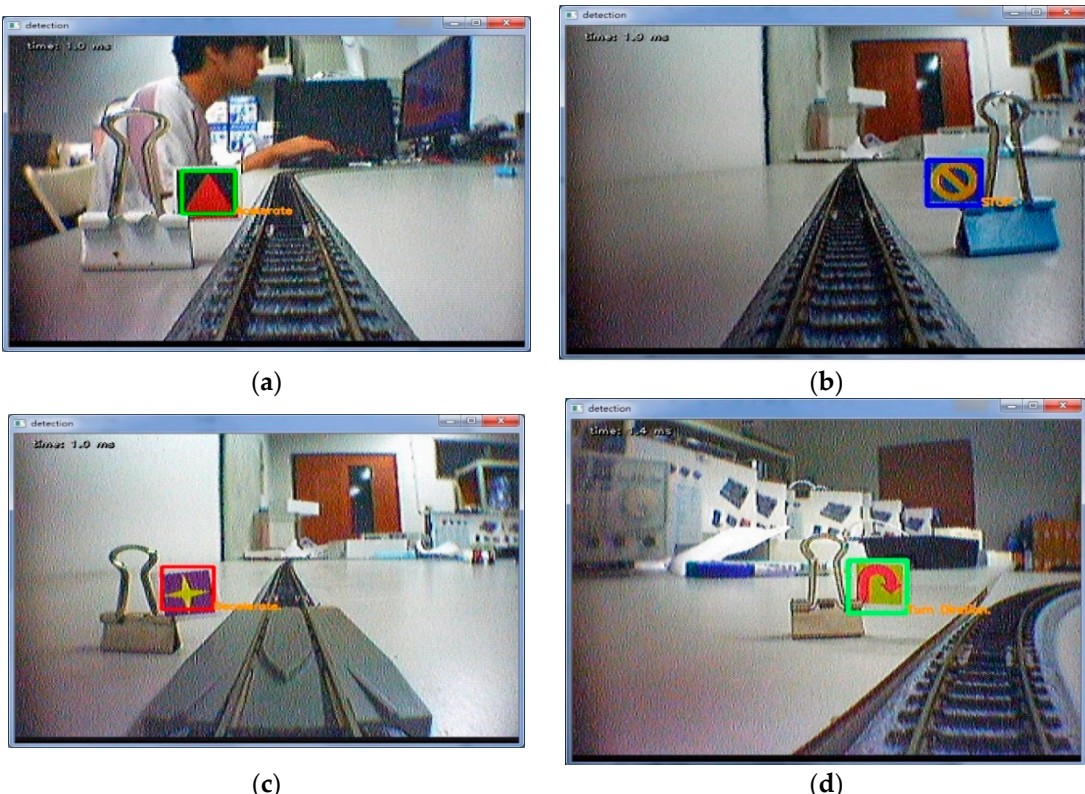

**Figure 9.** Confirmed detections of four predefined signs. (**a**) acceleration; (**b**) stop; (**c**) deceleration; (**d**) reverse.

*4.2. Manual Command Input Interface*

In addition, users can manually operate the motion of the model railcar through the command input interface. By clicking an "a" on the input interface, the power to drive the railcar will be increased by 10%. At the same time, three parameters N, PWM and V (voltage) will be displayed on the interface to show the current power setting of the model railcar. N is the ratio of current power setting to full power. For example, since the resolution of PWM is 8 bits, the PWM is encoded from 0 to 255. When N = 0.75, this means that 75% of available power is driving the model railcar (PWM = $255 \times 0.75 \cong 192$).

Similarly, the full power supply to drive the model railcar is 15V. Therefore, V = 11.25 also means the driving power is 75% ($15 \times 0.75 = 11.25$). When the power setting reaches to the maximum ratio of 0.95, continuing to click on "a" will not increase the speed of model railcar. Similarly, by clicking a "d" on the input interface, the power to drive the railcar will be decreased by 10%.

When the power setting reaches the minimum ratio of 0.35, keeping on clicking the "d" cannot further decrease the speed of model railcar. By clicking an "s" on the input interface, the model railcar will stop immediately. By clicking a "t" on the input interface, the model railcar will immediately reverse and also increase power by 10% to move away from original direction. Please refer to Figure 10 for above manual power setting operations.

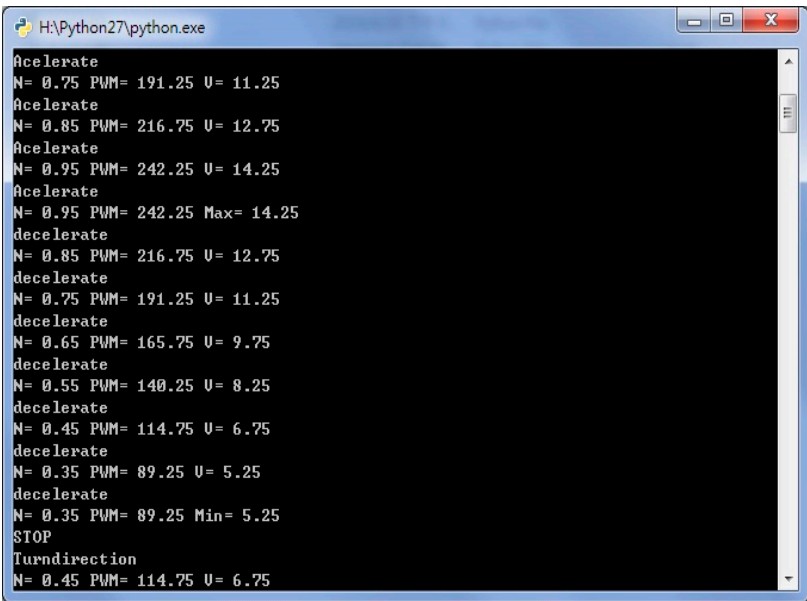

**Figure 10.** Manual power setting display windows.

### 4.3. Railcar Speed Estimation Experiment

Once the strong classifiers of four predefinded signs are trained, the user can assign different functions to them in different programs. In another experiment, the "reverse sign" in Figure 2 served as a positioning sign in an estimation of railcar speed. We evenly depolyed two positioning signs on the trackside of the rail loop with a length of 455 cm. Therefore, two positioning signs have even distances of 227.5 cm (0.002275 km) between them. When the railcar was operating, the program kept recording the times when the system detected the positioning sign. Dividing 227.5 cm by the time difference of two consecutive positioning sign detections, the system can get an estimation of railcar speed. The film of speed estimation experiment is available at https://www.youtube.com/watch?v=oLZXzY03Afc (accessed on 31 January 2019) [33]. The speed measurement function of railway CBTC system is simply achieved using the proposed method based on image recognition.

If we set up a counter of positioning sign detections in the program, multiplying 227.5 cm by the counter value can estimate the traveling distance of railcars.

## 5. Discussion

In this experiment, an integrated unmanned motion control system for railcars based on image recognition is built and verified successfully. In this paper, it is assumed that railcars operate in a fixed luminous environment. The surroundings have fixed ambient brightness like places within an indoor factory or a subway. In experience, the correct rate of image recognition is related to ambient brightness. Although histogram equalization is applied in the main program to normalize the brightness and increase the contrast of the images, when the railcar operates in brightness-varying outdoor surroundings, to better improve the correct rate we should train several strong classifiers in different conditions of ambient brightness. In implementation, the system can install an ambient light sensor to first detect the ambient brightness and then load the strong classifiers trained in the closest ambient brightness into the main program. Doing this may reduce the errors due to brightness varying. This can be done in future work.

We can also train more predefined signs to handle applications that are more practical in future work. For example, a railcar may roll over on rails with large curvature if the speed is too fast. To prevent this potential problem, one can put a warning sign ahead of the curved rail section.

When the system detects the warning sign, a warning message would show up on the display window to alert the operator. At the same time, the power automatically drops to 40% in order that

the railcar can pass this curved rail section safely at low speed. The camera perspective film of this preliminary experiment is available at https://www.youtube.com/watch?v=I4OEYRAn0Ak (accessed on 31 January 2019) [34].

Furthermore, like CBTC systems, the proposed system can estimate the speed of a railcar with the method introduced in Section 4.3. The optimal design of speed profiles that minimizes the energy consumption for a target running time can also be studied in the future. In addition, if we train more new predefined signs representing speed limits or speed commands and properly place them, a function for the prevention of speeding, or a function for speed-tracking can be carried out by extending the main application program. For example, a comprehensive reference table of speeds versus PWM power ratios and advanced motion control algorithms can be applied in the extended main program in future work.

## 6. Conclusions

In this research, an integrated unmanned motion control system of railcars based on image recognition is built and verified by experiment successfully. When this system detects a predefined sign, it can generate a new PWM power setting to drive the railcars within 0.3 of a second. The system response is very fast. Using a better computer can further improve the system performance. Since the sensing and control devices of the proposed system consists of a computer, camera and predefined signs only, both the implementation and maintenance costs are very low. Operating in surroundings of fixed ambient brightness, the possibility of malfunction of the proposed system is very small. Even if any malfunction does happen, it should be easy to find the cause and to fix it. Therefore, one contribution of this research is providing a cost affordable and reliable solution for motion control of unmanned railcars. In addition, many signaling systems are radio-based communication systems such as the CBTC system. They have electromagnetic compatibility (EMC) issues and easily suffer from electromagnetic interference (EMI) caused by lighting or intentional radio jamming launched by terrorists. Since cameras sense and convert light waves to images, and light waves are immune to electromagnetic interference, merging the proposed method of image recognition into signaling systems can improve safety at additional low cost. This is another contribution.

However, the main contribution of this research is to demonstrate the feasibility of graphic predefined signs serving as the commands for a computer-based vision system. In other words, the detections of different predefined signs can trigger different system responses to handle different tasks. With this concept and imagination, people can develop more versatile control systems based on computer vision. For example, the proposed method is also applicable to autonomous vehicles so that one can properly deploy predefined signs on the roadside to control the motions of autonomous vehicles using image recognition technology. Therefore, one can train more predefined signs as commands to accomplish motion control for more complicated tasks in future.

**Author Contributions:** Conceptualization, Y.-W.T. and R.-C.W.; methodology, Y.-W.T.; software, T.-W.H.; validation, T.-W.H., C.-L.P. and Y.-W.T.; formal analysis, Y.-W.T., C.-L.P.; investigation, Y.-W.T.; resources, R.-C.W.; data curation, T.-W.H.; writing—original draft preparation, Y.-W.T.; writing—review and editing, C.-L.P.; visualization, T.-W.H.; supervision, R.-C.W.; project administration, R.-C.W.; funding acquisition, R.-C.W. and Y.-W.T.

**Funding:** This research was funded by Ministry of Science and Technology, Taiwan, R.O.C., grant Nos. MOST 106-2221-E-214-008 and MOST 107-2622-E-214-003-CC3.

**Conflicts of Interest:** The authors declare no conflict of interest.

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
