# Peer review of "Motion Control System of Unmanned Railcars Based on Image Recognition"

_asi, doi:10.3390/asi2010009_

Round 1
Reviewer 1 Report
The paper presents a motion control system for unmanned railcars based on image recognition.
The paper is an interesting one. I appreciate the fact that there are not only presented the theoretical ideas for the system, but there are also being clearly shown the experimental setup, results and video of the full functional experiment.
As a suggestion for future work there could be considered more than 4 predefined signs for the motion commands: warning signs, different speed limits, etc.
Also, minor English language improvements can be done.
The conclusions should be extended for a better highlighting of authors contributions.
Author Response
Dear Reviewer,
The authors sincerely appreciate your effort and time to provide us your valuable comments and guidance to improve our paper .
Following your valuable comments and guidance, we have revised our paper. In the attached file, the details of the revisions are explained point by point and summarized in the section of Comments and Suggestions for Authors and Actions of Revisions for your further consideration.
Respectfully yours,
Yuan-Wei Tseng, Tsung-Wui Hung, Chung-Long Pan and Rong-Ching Wu
Comments and Suggestions for Authors and Actions of Revisions
The paper presents a motion control system for unmanned railcars based on image recognition.
The paper is an interesting one. I appreciate the fact that there are not only presented the theoretical ideas for the system, but there are also being clearly shown the experimental setup, results and video of the full functional experiment.
Point 1: As a suggestion for future work there could be considered more than 4 predefined signs for the motion commands: warning signs, different speed limits, etc.
Response 1: Discussion section has been extend to discuss the potential of adding more predefined signs to improve the versatility of the proposed system to handle warning, speeding prevention and speed tracking tasks in future work.
Point 2: Also, minor English language improvements can be done.
Response 2: One of the author is an US Citizen, he have fixed typos and grammatical errors as many as possible.
Point 3: The conclusions should be extended for a better highlighting of authors contributions.
Response 3: Conclusions section has been expanded to include the main results and numbers of experiment. Contributions are also better highlighted.
Reviewer 2 Report
The manuscript titled “Motion Control System of Unmanned Railcars Based on Image Recognition” represents interesting scientific research. The idea of the research is interesting and present enough novelty. The manuscript should attract an audience in the processing Image recognition, Computer vision and Motion control.
The manuscript title is accurate and concise. In the entire manuscript, authors use standard technical and scientific terminology. After Introduction, the authors explained in detailed, used image recognition methodology. Simulations and results were conducted according to the scientifically correct approach. The conclusions are logical and based on the results of the research. The manuscript topics fit in Applied System Innovation aims and scope, especially in Smart electromechanical system, as well as Mathematical control system design. I recommended this manuscript to be accepted after major revisions.
Comments for authors:
1. Please slightly expanded the abstract. A) Summarise the article's main findings; B) Indicate the main conclusions or interpretations. The abstract should be an objective representation of the article.
2. Please remove figures 1 and 2 from Introduction and move it to the next section. Or move it to the newly added section (e.g. “System design” or something else). Please, add more adequate references in the Introduction section.
3. Suggest expanding the Discussion section.
4. Slightly expand the Conclusions section with the main results and numbers.
5. Use MDPI standard font (Palatino Linotype) on figures if you can.
6. Figure captions must finish with dot at the end (e.g. line 43, etc.).
7. The variable names must have the same font style and size in equations, on figures, tables and in the manuscript text. Please describe/introduce all variables used in equations or on figures in the manuscript text.
8. All equations must be adequately cited in the entire manuscript.
9. Replace figures 6 and 7. Please, do not use folder screenshot as a figure.
10. Please introduce abbreviations if you want to use it in the further manuscript text (e.g. RGB, GPS, PWM, CMOS, WLAN, etc.).
11. Avoid using one sentence paragraph (e.g. line 103, etc.).
12. Please, double check all the references and reference style.
13. In the manuscript text has some typos (e.g. line 160 – “x,y” -> “x, y”, line 202 – “jth” -> “j-th” , etc.).
Author Response
Dear Reviewer,
The authors sincerely appreciate your effort and time to provide us your valuable comments and guidance to improve our paper.
Following your valuable comments and guidance, we have revised our paper. In the attached file, the details of the revisions are explained point by point and summarized in the section of Comments for Authors and Actions of Revisions for your further consideration.
Respectfully yours,
Yuan-Wei Tseng, Tsung-Wui Hung, Chung-Long Pan and Rong-Ching Wu
Comments and Suggestions for Authors
The manuscript titled “Motion Control System of Unmanned Railcars Based on Image Recognition” represents interesting scientific research. The idea of the research is interesting and present enough novelty. The manuscript should attract an audience in the processing Image recognition, Computer vision and Motion control.
The manuscript title is accurate and concise. In the entire manuscript, authors use standard technical and scientific terminology. After Introduction, the authors explained in detailed, used image recognition methodology. Simulations and results were conducted according to the scientifically correct approach. The conclusions are logical and based on the results of the research. The manuscript topics fit in Applied System Innovation aims and scope, especially in Smart electromechanical system, as well as Mathematical control system design. I recommended this manuscript to be accepted after major revisions.
Comments for authors and Actions of Revisions
Point 1: Please slightly expanded the abstract. A) Summarize the article's main findings; B) Indicate the main conclusions or interpretations. The abstract should be an objective representation of the article.
Response 1: abstract section has been revised based on reviewer’s valuable comments to include the objectives of this paper, main findings in experiments and the potential of merging the proposed system into current CBTC train control system to improve safety.
Point 2: Please remove figures 1 and 2 from Introduction and move it to the next section. Or move it to the newly added section (e.g. “System design” or something else). Please, add more adequate references in the Introduction section.
Response 2: Since Figures 1 and 2 are system diagrams of TBTC and CBTC systems discussed in introduction, we moved Figures 1 and 2 to the end of introduction for better visualization while keeping the integrity of the introduction section.
Point 3: Suggest expanding the Discussion section.
Response 3: Discussion section has been extend to discuss the potential of adding more predefined signs to improve the versatility of the proposed system to handle warning, speeding prevention and speed tracking tasks in future work.
Point 4: Slightly expand the Conclusions section with the main results and numbers.
Response 4: Conclusions section has been expanded to include the main results and numbers of experiment. Contributions are also better highlighted.
Point 5: Use MDPI standard font (Palatino Linotype) on figures if you can.
Response 5: Captions of all figures have been revised to MDPI standard font. However, some texts inside the figures are not editable and have to leave them as-is.
Point 6: Figure captions must finish with dot at the end (e.g. line 43, etc.).
Response 6: Figure captions all finish with dot at the end now.
Point 7: The variable names must have the same font style and size in equations, on figures, tables and in the manuscript text. Please describe/introduce all variables used in equations or on figures in the manuscript text.
Response 7: Comprehensive checks have done to make sure that variable names have the same font style and size in equations and in the manuscript text as many as possible. To authors’ best knowledge, all variables are now properly described in this paper after revisions.
Point 8: All equations must be adequately cited in the entire manuscript.
Response 8: All equations are now being cited in the entire manuscript after
revisions.
Point 9: Replace figures 6 and 7. Please, do not use folder screenshot as a figure.
Response 9: Figures 6 and 7 have been replaced by new figures of better quality.
Point 10: Please introduce abbreviations if you want to use it in the further manuscript text (e.g. RGB, GPS, PWM, CMOS, WLAN, etc.).
Response 10: The original words of all abbreviations are now included in the revised manuscript text.
Point 11: Avoid using one sentence paragraph (e.g. line 103, etc.).
Response 11: There is no more one-sentence paragraph in the revised
manuscript text.
Point 12: Please, double check all the references and reference style.
Response 12: All references are revised to comply with the reference style of ASI-template.
Point 13: In the manuscript text has some typos (e.g. line 160 – “x,y” -> “x, y”,
line 202 – “jth” -> “j-th” , etc.).
Response 13: To authors’ best knowledge, all typos have been corrected after revisions.
Round 2
Reviewer 2 Report
The authors have addressed all the reviewers' comments, and the manuscript in its current version is improved compared to the original.
I have no further comments, and the revised manuscript can be accepted.